# Structure of the GOLD-domain seven-transmembrane helix protein family member TMEM87A

**Christopher M Hoel**[1,2,3†], **Lin Zhang**[1,2,3†], **Stephen G Brohawn**[1,2,3*]

[1]Department of Molecular and Cell Biology, University of California, Berkeley, Berkeley, United States; [2]Helen Wills Neuroscience Institute, University of California, Berkeley, Berkeley, United States; [3]California Institute for Quantitative Biology (QB3), University of California, Berkeley, Berkeley, United States

**\*For correspondence:**
brohawn@berkeley.edu

[†]These authors contributed equally to this work

**Competing interest:** The authors declare that no competing interests exist.

**Abstract** TMEM87s are eukaryotic transmembrane proteins with two members (TMEM87A and TMEM87B) in humans. TMEM87s have proposed roles in protein transport to and from the Golgi, as mechanosensitive ion channels, and in developmental signaling. TMEM87 disruption has been implicated in cancers and developmental disorders. To better understand TMEM87 structure and function, we determined a cryo-EM structure of human TMEM87A in lipid nanodiscs. TMEM87A consists of a Golgi-dynamics (GOLD) domain atop a membrane-spanning seven-transmembrane helix domain with a large cavity open to solution and the membrane outer leaflet. Structural and functional analyses suggest TMEM87A may not function as an ion channel or G-protein coupled receptor. We find TMEM87A shares its characteristic domain arrangement with seven other proteins in humans; three that had been identified as evolutionary related (TMEM87B, GPR107, and GPR108) and four previously unrecognized homologs (GPR180, TMEM145, TMEM181, and WLS). Among these structurally related GOLD domain seven-transmembrane helix (GOST) proteins, WLS is best characterized as a membrane trafficking and secretion chaperone for lipidated Wnt signaling proteins. We find key structural determinants for WLS function are conserved in TMEM87A. We propose TMEM87A and structurally homologous GOST proteins could serve a common role in trafficking membrane-associated cargo.

## Editor's evaluation

Your work addresses the mechanisms of action of the transmembrane proteins TMEM87A and TMEM87B, which are thought to play a role in protein transport but have been implicated in other processes as well, such as signaling and acting as mechanosensitive ion channels. The study represents an important advance in the understanding of this poorly characterized family of proteins.

## Introduction

TMEM87 proteins are transmembrane proteins found in eukaryotic organisms from fungi and plants to mammals (*Edgar, 2007*). In humans, two paralogs have been identified: TMEM87A and TMEM87B (*Edgar, 2007*). TMEM87A and TMEM87B have been associated with protein transport to and from the Golgi, mechanosensitive cation channel activity, and cardiac and other developmental processes (*Hirata et al., 2015*; *Shin et al., 2020*; *Russell et al., 2014*; *Patkunarajah et al., 2020*). In humans, TMEM87 proteins have been implicated in developmental diseases and cancers (*Cooper et al., 2020*; *Shaver et al., 2016*; *Yu et al., 2016*; *Digilio et al., 2022*).

Several lines of evidence support a role for TMEM87 in protein trafficking. First, TMEM87s localize to the Golgi (*Hirata et al., 2015*; *Shin et al., 2020*). Second, overexpression of TMEM87A or TMEM87B partially rescues defective endosome-to-*trans*-Golgi network retrograde traffic in HEK293 cells lacking the Golgi-associated retrograde protein (GARP) complex member VPS54 (*Hirata et al., 2015*). Third, TMEM87A has been identified as retrograde cargo captured by mitochondrial golgins (*Shin et al., 2020*). A recent study, however, associated TMEM87A with mechanosensitive cation channel activity (*Patkunarajah et al., 2020*). Deflection of micropillar culture supports resulted in cationic currents in melanoma cells that were reduced by TMEM87A expression knockdown. Additionally, TMEM87A overexpression resulted in mechanosensitive currents in PIEZO1 knockout HEK293T cells (*Patkunarajah et al., 2020*). TMEM87A knockout melanoma cells show increased adhesion strength and decreased migration, suggesting potential roles for TMEM87A in these processes in cancers (*Patkunarajah et al., 2020*).

TMEM87B has been linked to developmental processes in recurrent 2q13 microdeletion syndrome (*Russell et al., 2014*; *Yu et al., 2016*; *Digilio et al., 2022*). Patients with this syndrome exhibit cardiac defects, craniofacial anomalies, and developmental delays. The vast majority of 2q13 microdeletions include TMEM87B (*Yu et al., 2016*). In zebrafish, morpholino knockdown of TMEM87B resulted in cardiac hypoplasia (*Russell et al., 2014*). In a patient with a severe cardiac phenotype, whole-exome sequencing uncovered a paternally inherited TMEM87B missense mutation and chromosome 2 deletion inherited from an unaffected mother (*Yu et al., 2016*). Both TMEM87A and TMEM87B have additionally been implicated in cancers such as non-small cell lung cancer through fusion or suspected interaction with oncogenes (*Cooper et al., 2020*; *Shaver et al., 2016*; *Li et al., 2008*).

Bioinformatic analysis has grouped TMEM87A and TMEM87B in a small family of proteins (termed lung 7TM receptors or LUSTRs) with so-called orphan GPCRs GPR107 and GPR108 (*Edgar, 2007*). Like TMEM87s, GPR107, and GPR108 localize to the Golgi and have suggested roles in protein trafficking. GPR107 is implicated in the transport of *Pseudomonas aeruginosa* Exotoxin A, *Campylobacter jejuni* cytolethal distending toxin (CDT), and ricin, while GPR107 knockout cells display deficits in receptor-mediated endocytosis and recycling (*Tafesse et al., 2014*; *Carette et al., 2011*; *Elling et al., 2011*; *Zhou et al., 2014*). GPR108 is critical for the transduction of a majority of AAV serotypes (*Meisen et al., 2020*; *Dudek et al., 2020*). While a GPR107 knockout mouse was embryonic lethal, a GPR108 knockout mouse was viable and revealed a potential role for GPR108 in the regulation of Toll-like receptor-mediated signaling (*Zhou et al., 2014*; *Dong et al., 2018*).

Despite being implicated in important cell biological processes and human health, our molecular understanding of TMEM87s and LUSTR proteins is limited and no experimental structures of these proteins have been reported to date. Here, we present a cryo-electron microscopy (cryo-EM) structure of human TMEM87A. Through structural and bioinformatic analyses, we find TMEM87A belongs to a broader protein family we term GOLD-domain seven transmembrane (GOST) proteins. We speculate TMEM87A and other GOST proteins could function as trafficking chaperones for membrane-associated cargo.

## Results

Full-length human TMEM87A was expressed and purified from Sf9 insect cells and reconstituted into lipid nanodiscs composed of MSP1D1 and a mixture of DOPE, POPS, and POPC lipids (*Figure 1—figure supplement 1*). Cryo-EM was used to determine the structure of TMEM87A to a nominal resolution of 4.7 Å, with better resolved regions reaching ~4.3 Å in the core of the protein. The map was of sufficient quality to place secondary structure elements unambiguously and, using an AlphaFold2 predicted structure as a starting model, to place and refine 395/555 residues (*Figure 1*, *Figure 1—figure supplements 2 and 3*, *Supplementary file 1*). The modeled portion of TMEM87A and corresponding predicted structure are similar (overall r.m.s.d.=1.8 Å) with a minor difference in the relative position of extracellular and transmembrane regions. About 37 N-terminal residues (amino acids 1–37), 88 C-terminal residues (475-555), and 36 residues in loops within the extracellular domain (145–173, 192–198) were not resolved in the cryo-EM map and are unmodeled.

TMEM87A is composed of an extracellular beta-sandwich domain atop a G-protein coupled receptor (GPCR)-like seven-transmembrane (7TM) domain (*Figure 1*). The extracellular beta-sandwich domain is formed by seven beta strands in two opposing sheets, with three strands on the N-terminal face and four on the opposing face. The N-terminal face consists of β1, β3, β7, and a region between

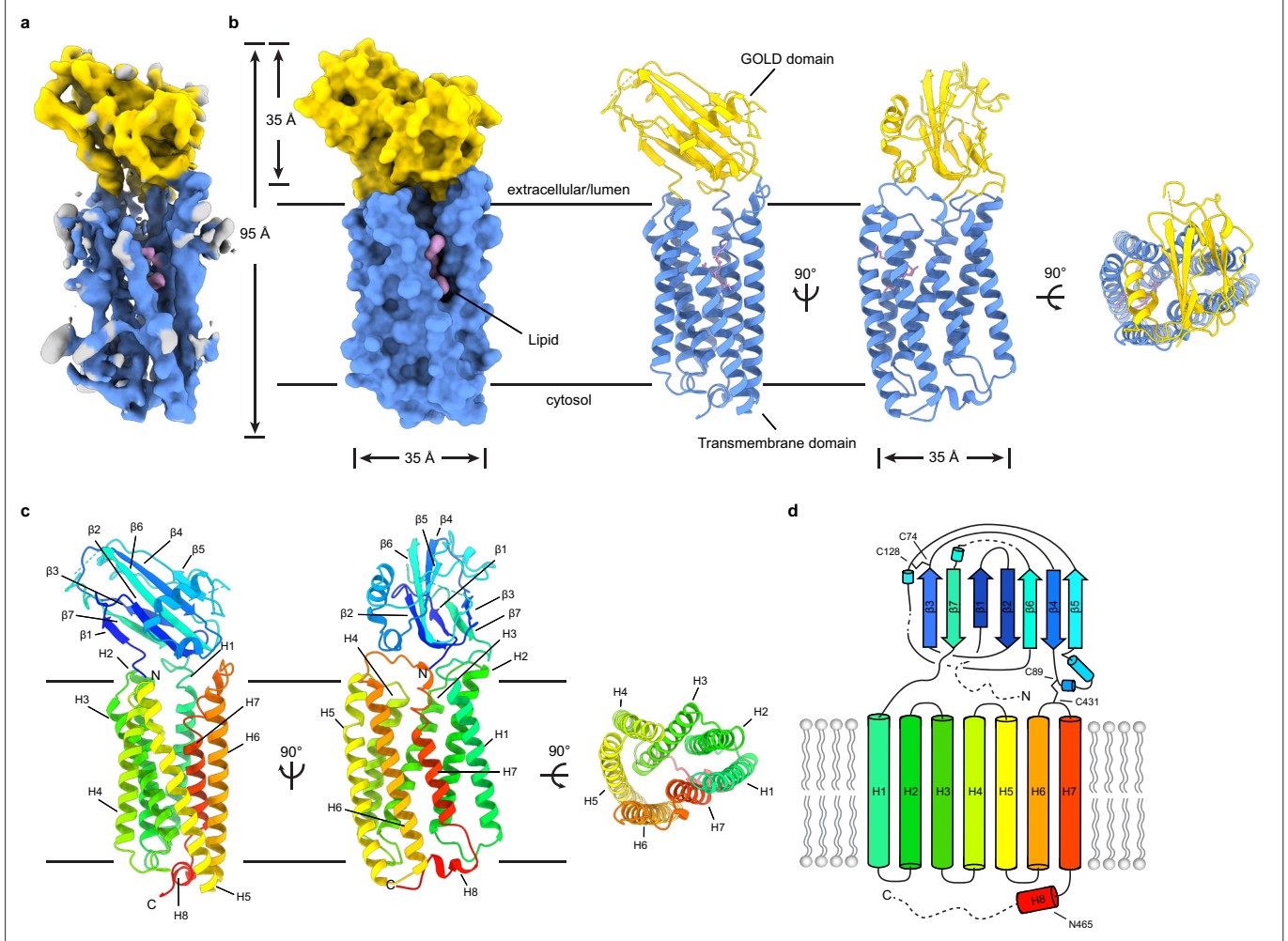

**Figure 1.** Structure of TMEM87A in lipid nanodiscs. (**a**) 4.7 Å cryo-EM map and (**b**) model for TMEM87A viewed from the plane of the membrane and the extracellular or lumenal side (right). The GOLD domain is colored yellow, seven-transmembrane domain colored blue, and modeled phospholipid colored pink. (**c**) TMEM87A with rainbow coloring from N-terminus (blue) to C-terminus (red). (**d**) Corresponding cartoon of domain topology with rainbow coloring from N-terminus (blue) to C-terminus (red). Positions of disulfide bonds and residues noted in the text are indicated. EM, electron microscopy.

The online version of this article includes the following figure supplement(s) for figure 1:

**Figure supplement 1.** TMEM87A biochemistry.

**Figure supplement 2.** Cryo-EM data, processing, and validation.

**Figure supplement 3.** Cryo-EM data processing pipeline.

β5 and β6 lacking secondary structure that packs against β3 at the top of the sheet. The opposing face of the β-sandwich is formed from β2, β4, β5, and β6 with a helix-turn-helix motif between β4 and β5. A short 10 residue loop extends from the end of β7 to the 7TM domain at the extracellular side of H1. Two disulfide bonds are observed. The first, between C74 and C128, connects two loops at the top of this extracellular domain. The second, between C89 and C431, tethers the bottom of the extracellular domain to the top of the 7TM domain in the H6–H7 linker, perhaps constraining their relative movement.

The transmembrane region of TMEM87A is structurally similar to other 7TM proteins, with a few notable differences (*Figure 2*, *Figure 2—figure supplement 1*). Using Dali to compare the isolated TMEM87A transmembrane domain to all experimentally determined protein structures returns 7TM proteins as clear structural homologs, with the fungal class D GPCR Ste2, microbial opsins, and

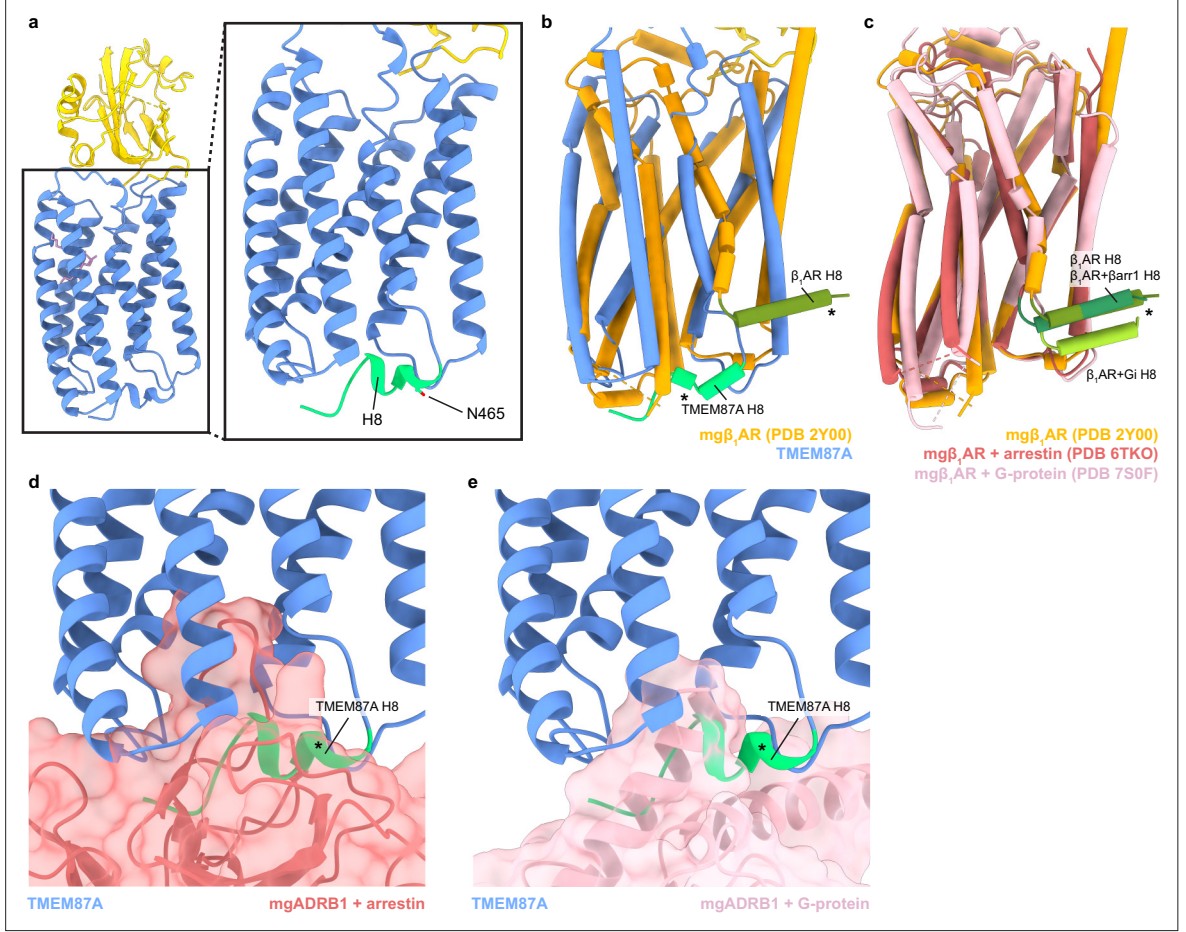

**Figure 2.** The TMEM87A transmembrane domain and comparison to an A class GPCR. (**a**) View of the TMEM87A model from the 'back' in the plane of the membrane and zoomed view highlighting the position of helix 8 with residue N465 indicated (corresponding to TMEM87B N456D implicated 2q13 deletion syndrome). (**b**) Overlay of TMEM87A (blue) and β1 adrenergic receptor (mgβ1AR PDB: 2Y00, orange) transmembrane domains. (**c**) Overlay of mgβ1AR alone (PDB: 2Y00, orange), in complex with arrestin (PDB: 6TKO, red) in complex with heterotrimeric Gi-proteins (PDB: 7S0F, pink). (**d, e**) Overlay of TMEM87A (blue) and (**d**) arrestin (red surface) or (**e**) heterotrimeric Gi (pink surface) from mgβ1AR complex structures(PDB: 6TKO, 7S0F). Helix 8s (H8s) are colored shades of green and denoted with asterisks.

The online version of this article includes the following figure supplement(s) for figure 2:

**Figure supplement 1.** Comparison of TMEM87A and GPCR structures.

**Figure supplement 2.** TMEM87A proteoliposome recordings.

**Figure supplement 3.** Conservation of TMEM87A and WLS.

additional class A GPCRs among the hits (*Holm, 2020*; *Velazhahan et al., 2021*; *Hayashi et al., 2020*; *Yamamoto et al., 2009*).

Since TMEM87A shares structural features with ion-conducting opsins and previous work implicated TMEM87A in mechanosensitive cation conduction (*Patkunarajah et al., 2020*), we asked whether TMEM87A displayed channel activity in isolation. We purified and reconstituted TMEM87A into proteoliposomes and recorded currents across patched membranes in response to membrane stretch induced by pressure steps. We observed neither basal nor mechanically activated currents in TMEM87A reconstituted proteoliposomes, in contrast to the mechanosensitive ion channel TRAAK used as a positive control (*Figure 2—figure supplement 2*). This result is consistent with the lack of a clear ion conducting path in the TMEM87A structure. We conclude that under these conditions, TMEM87A does not form a mechanosensitive ion channel.

Since TMEM87A contains a 7TM domain, we next asked whether it has structural features consistent with G-protein coupled receptors. TMEM87A and Class A GPCRs have a similar transmembrane helix organization, for example, TMEM87A and the β1-adrenergic receptor are superimposed with

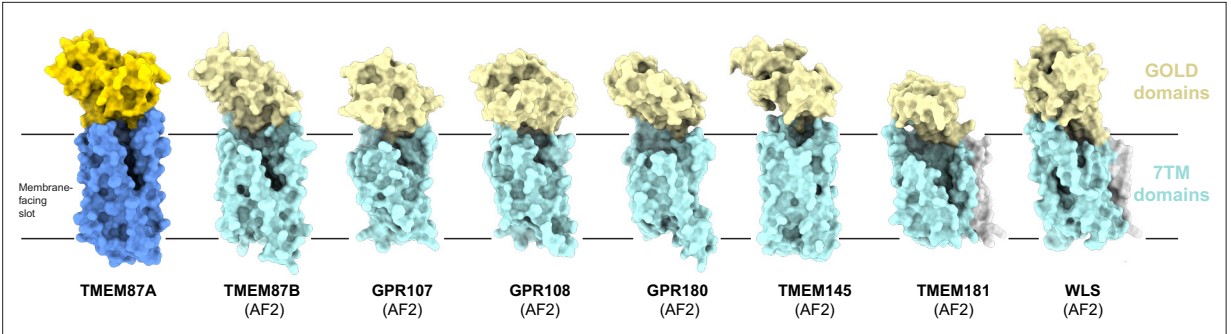

**Figure 3.** Predicted structural homology of GOST (GOLD domain seven transmembrane helix) family proteins. Experimentally determined TMEM87A structure and AlphaFold2 predicted structures of all identified human GOST family proteins (TMEM87B, GPR107, GPR108, GPR180, TMEM145, TMEM181, and WLS) shown as surfaces in the same orientation from the membrane plane. GOLD domains are yellow, seven-transmembrane domains are blue, N-terminal helices (if present) are gray, and membrane-facing slots are outlined in pink. Low confidence (<80 pLDDT) regions of predicted structures are removed for clarity (full structures are displayed in *Figure 3—figure supplement 2*).

The online version of this article includes the following figure supplement(s) for figure 3:

**Figure supplement 1.** GOLD domain comparisons.

**Figure supplement 2.** AlphaFold2 predictions of GOST family protein structures.

an overall r.m.s.d. of 4.13 Å (*Figure 2*). Intriguingly, the position of cytoplasmic helix 8 in TMEM87A differs substantially from experimental structures of GPCRs. In TMEM87A, helix 8 turns back towards the center of the protein and packs against the bottom of the transmembrane domain (*Figure 2a and b*). In experimental GPCR structures, helix 8 is instead rotated nearly 180° and adopts a different position projecting toward the surrounding membrane (*Warne et al., 2011*; *Lee et al., 2020*; *Alegre et al., 2021*; *Figure 2b–c*). This position of helix 8 in GPCRs appears necessary to accommodate G-protein or arrestin binding and is similarly positioned in apo- and complexed-receptor structures (*Figure 2c*). The position of helix 8 in TMEM87A would sterically clash with putative G-protein or arrestin binding through interfaces analogous to those observed in GPCR structures (*Figure 2c–e*). Both helix 8 and the cytoplasmic surface it interacts with are highly conserved in TMEM87A across species (*Figure 2—figure supplement 3*). Notably, the TMEM87B point mutation implicated in 2q13 deletion syndrome (TMEM87B N456D) corresponds to TMEM87A residue N465 at the TM7-helix 8 junction (*Yu et al., 2016*; *Figure 2a*), suggesting helix 8 is important for TMEM87B function. In addition to this difference in helix 8 position, TMEM87A does not share canonical motifs of class A or D GPCRs including the class A NPxxY activation motif, PIF motif, and D(E)/RY motif or the class D LPLSSMWA activation motif (*Velazahhan et al., 2021*; *Palczewski et al., 2000*; *Wang et al., 2020*). Taken together, these differences suggest that TMEM87A may not couple to G proteins or arrestins like canonical GPCRs.

We found the extracellular/lumenal region of TMEM87A adopts a Golgi dynamics (GOLD) domain fold (*Figure 3*, *Figure 3—figure supplement 1*). Using Dali to compare the isolated TMEM87A extracellular β-sandwich domain (residues 38–218) to all experimentally determined protein structures returns human p24 GOLD domain proteins as top hits. Superposition of the TMEM87A extracellular domain and p24 shows conservation of β strand topology throughout the domain, with differences predominantly in regions connecting β strands including the helix-turn-helix motif in the TMEM87A GOLD domain (*Holm, 2020*; *Nagae et al., 2016*; *Figure 3—figure supplement 1a–c*). GOLD domain proteins have established roles in the secretory pathway (*Nagae et al., 2016*; *Anantharaman and Aravind, 2002*) and p24 proteins are implicated as cargo receptors for protein transport, with the GOLD domain mediating cargo recognition (*Pastor-Cantizano et al., 2016*; *Mendes and Costa-Filho, 2022*). Among different p24 proteins, the GOLD domain loops are among the most variable regions and have been hypothesized to recognize different cargo (*Nagae et al., 2016*). Identification of a GOLD domain in TMEM87A suggests a structural explanation for the proposed roles of TMEM87s in protein transport, with major differences in GOLD domain loops between TMEM87A and p24 perhaps indicative of differences in interacting partners.

We next asked whether further structural comparison could provide insight into TMEM87A functions. We performed a Dali search against the entire AlphaFold2-predicted human proteome using

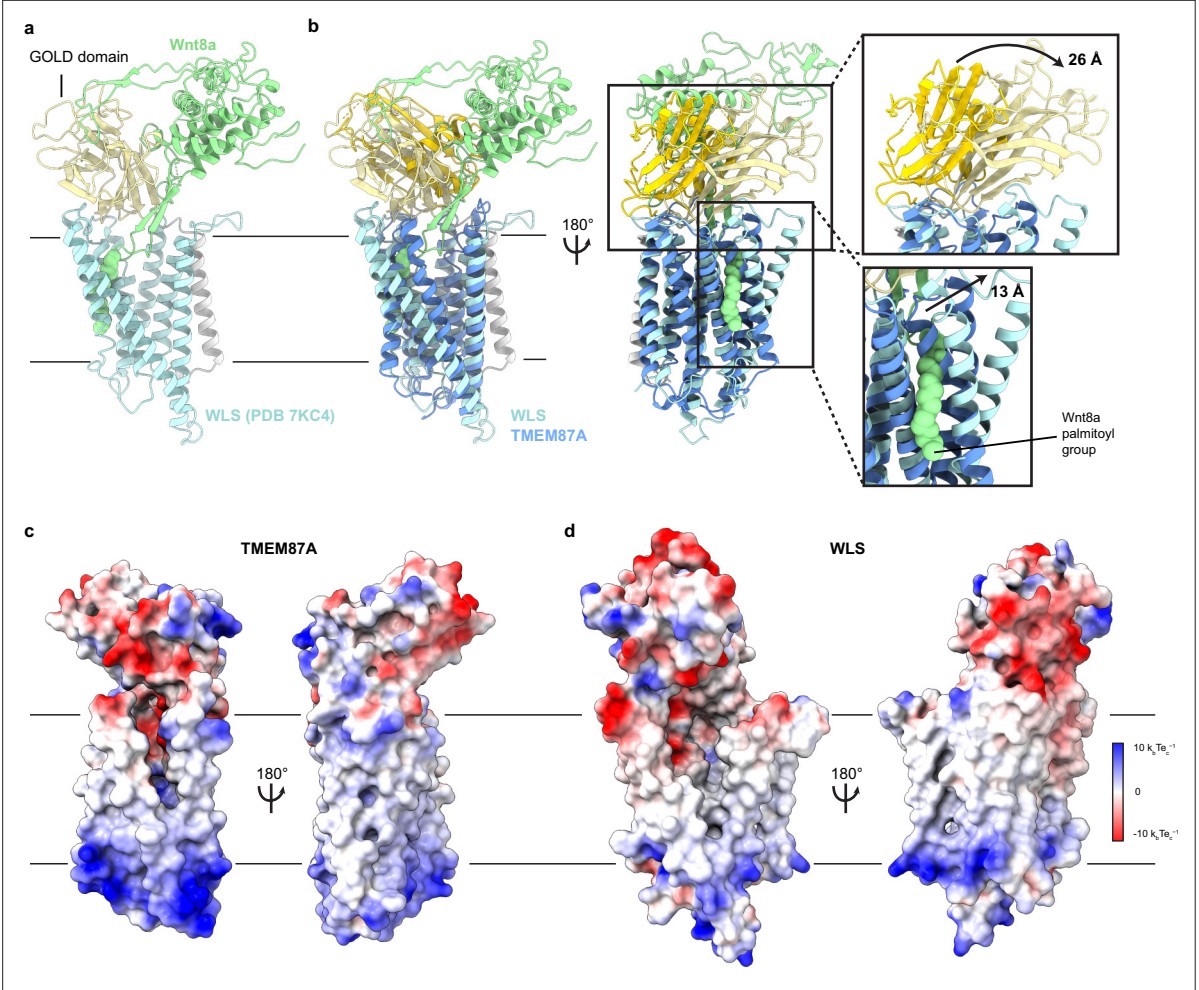

**Figure 4.** Structural comparison of TMEM87A and WLS bound to Wnt. (**a**) WLS-Wnt8a complex structure (PDB: 7KC4) with WLS GOLD domain light yellow, the WLS seven-transmembrane domain light blue, the N terminal helix light gray, and Wnt8a light green. (**b**) Overlay of TMEM87A and WLS (PDB 7KC4) experimental structures, with TMEM87A GOLD domain colored yellow and TMEM87A transmembrane domain colored blue. Zoomed in views highlight the differences in relative domain orientation between TMEM87A and WLS. (**c, d**) Views of (**c**) TMEM87A and (**d**) WLS electrostatic surfaces. Scale is from −10 $k_b Te_c^{-1}$ to 10 $k_b Te_c^{-1}$.

the experimental TMEM87A structure as the reference (*Holm, 2020*; *Jumper et al., 2021*; *Varadi et al., 2022*). This search uncovered proteins predicted to have the same domain topology consisting of a GOLD domain fused to a 7TM domain: all four members of the LUSTR family and four previously unrecognized homologs GPR180, TMEM145, TMEM181, and Wntless (WLS) (*Figure 3*, *Figure 3— figure supplement 1d–j*, *Figure 3—figure supplement 2a–g*). The predicted structures are all structurally homologous to TMEM87A (overall r.m.s.d range from 1.85 to 3.87 Å). We propose an expansion of the LUSTR family to include these additional members and to name them GOST proteins (for GOLD domain seven-transmembrane helix proteins).

While three newly identified GOST proteins have been minimally characterized to date (prior work has associated GPR180 with TGF-B signaling and TMEM181 with *Escherichia coli* CDT toxicity *Balazova et al., 2021*; *Carette et al., 2009*), WLS (also known as GPR177 or evenness interrupted homolog [EVI]) has been extensively studied. WLS plays an essential role in chaperoning lipidated and secreted Wnt signaling proteins between internal membranes and the cell surface (*Bänziger et al., 2006*; *Bartscherer et al., 2006*; *Fu et al., 2009*). Recent cryo-EM structures of WLS bound to Wnt3a or Wnt8a provide insight into the basis for chaperone-client interactions (*Nygaard et al., 2021*; *Zhong et al., 2021*; *Figure 4a*). The extracellular region of Wnt interacts extensively with the WLS GOLD domain and an extended palmitoleated hairpin of Wnt that is buried in the large hydrophobic cavity

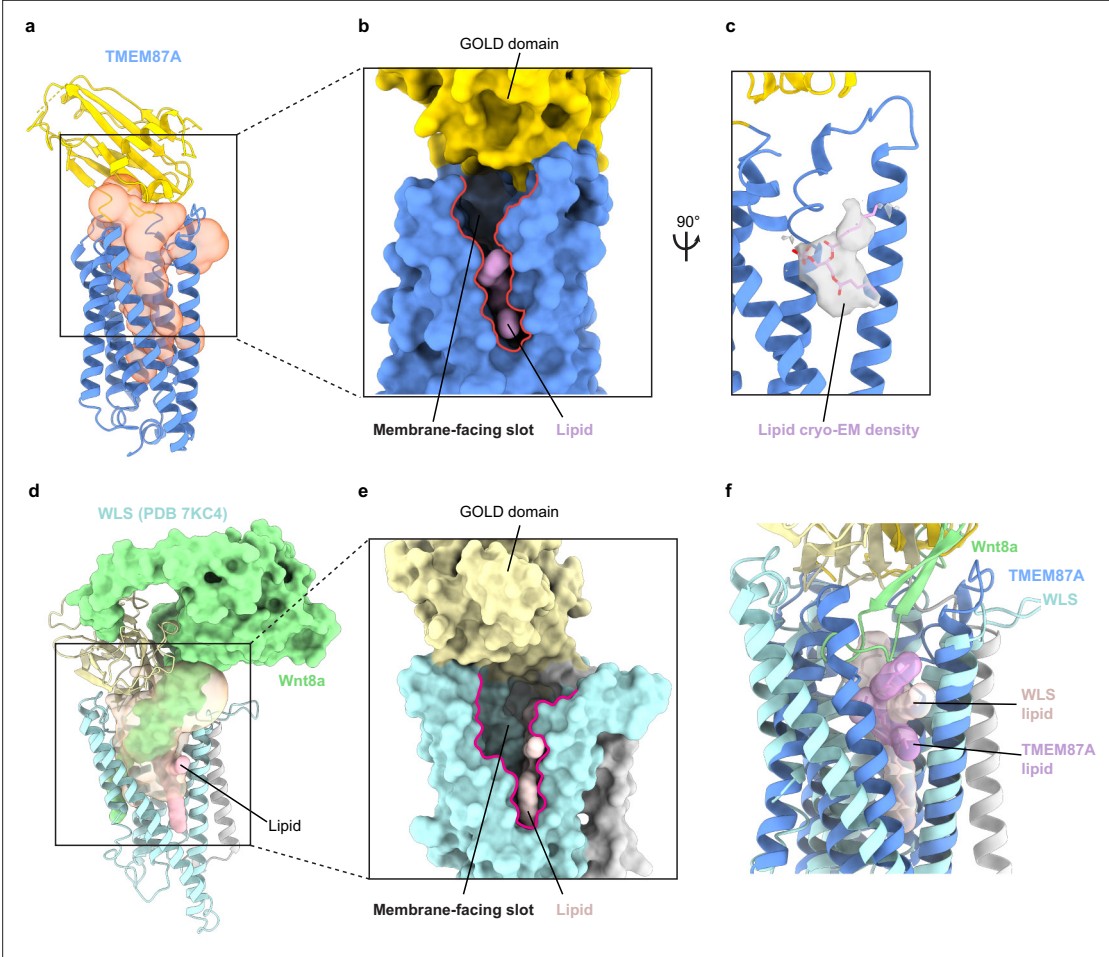

**Figure 5.** TMEM87A and WLS cavities are exposed to solution and membrane. (**a**) TMEM87A structure with CASTp calculated internal cavity shown as a transparent orange surface. (**b**) Surface of TMEM87A with membrane-facing slot (red outline) connecting the cavity and upper leaflet of the bilayer highlighted. (**c**) Modeled phospholipid (pink) is shown as a surface (left) or within cryo-EM density (right). (**d**) WLS (PDB: 7KC4) structure with CASTp calculated internal cavity shown as a transparent yellow surface, with bound Wnt8a (pale green surface) and phospholipid (pink) shown filling the cavity. (**e**) Zoomed in view shows surface representation of WLS (PDB: 7KC4) with open membrane-facing slot (red outline) and phospholipid (pink). (**f**) Overlay of the experimental TMEM87A and WLS structures showing relative positions of the H6 membrane-facing slot and modeled phospholipids. EM, electron microscopy.

within the WLS 7TM domain. Comparison of TMEM87A with WLS shows that the transmembrane regions are well aligned with two notable differences. First, WLS H4 and H5 are shifted approximately 13 Å away from the center of the protein, resulting in a larger cavity compared to TMEM87A necessary to accommodate the insertion of the lipidated Wnt hairpin (*Figure 4b*). The relative orientation of the GOLD and 7TM domains in TMEM87A and WLS also differs, with the WLS GOLD domain shifted approximately 26 Å in the same direction of movement as the H4/H5 shift (*Figure 4b*). Intriguingly, TMEM87A and WLS exhibit similar overall patterns of surface electrostatic charge including in the 7TM cavity that, in WLS, mediates binding of the Wnt lipidated hairpin (*Figure 4c–d*).

Perhaps the most striking structural feature of TMEM87A is the large cavity within the 7TM domain, a feature that is similar in several respects to the cavity in WLS structures (*Figure 5a and d*). In TMEM87A, this cavity measures approximately 8,637 Å (*Shin et al., 2020*) as compared to 12,350 Å (*Shin et al., 2020*) for the WLS cavity. It is funnel-shaped, presenting a large opening to the extracellular side near the GOLD domain and tapering closed near the intracellular surface above helix 8. The corresponding cavity in experimental WLS structures is filled by the lipidated Wnt3a/8 a hairpin (*Figure 5d*). Interestingly, the predicted WLS structure shows a smaller cavity relative to experimental WLS-Wnt structures. This suggests the apo-WLS cavity could dilate to accommodate Wnt binding (*Figure 3—figure supplement 2h–j*). The TMEM87A cavity is also exposed to the lipid membrane. H5

and H6 are splayed apart to open a large 'slot' between the cavity and the upper leaflet of the bilayer (*Figure 5b*). Density within the membrane exposed portion of the hydrophobic cavity is consistent with a bound phospholipid. A partial lipid is modeled based on density, chemical environment, and relationship to WLS, but we note this assignment is speculative given the relatively low resolution of the reconstruction. Notably, the WLS cavity is open to the upper leaflet through a similar slot between H5 and H6 and a phospholipid is bound in a similar position in experimental structures (*Figure 5d–f*). The slot in WLS is presumably important because it permits access of the lipidated Wnt hairpin from the membrane into the cavity binding site. Mapping sequence conservation onto TMEM87A and WLS structures shows a similar pattern with strongly conserved residues in the trans-membrane region lining the internal cavity (*Figure 2—figure supplement 3*). Taken together, the structural homology between TMEM87A and WLS suggests they could similarly have evolved to bind and facilitate membrane trafficking of hydrophobic cargo.

## Discussion

In this study, we report the first experimental structure of TMEM87A and consider its potential functional roles. TMEM87A shows general structural homology to ion transporting or conducting opsins and a recent report proposed TMEM87A contributes to mechanosensitive ion channel activity in cultured cells (*Patkunarajah et al., 2020*). We found no evidence of channel activity from purified TMEM87A reconstituted into proteoliposomes. This observation is consistent with the structure because while TMEM87A contains a solvent exposed cavity in its 7TM domain, it is sealed to the cytoplasmic side and open to the lipid bilayer, unlike typical ion conduction pathways. We cannot exclude the possibility that TMEM87A contributes to mechanosensitive ion channel currents under other conditions, in complex with other proteins, or in a cellular context. Still, further structural analyses and relationship to WLS suggest it may rather act indirectly to impact ion channel function, perhaps through regulating transport, expression, or activity of a mechanosensitive ion channel.

We discovered that TMEM87 consists of an extracellular or lumenal G̲O̲L̲D domain atop a membrane-spanning s̲even-t̲ransmembrane helix domain and shares this characteristic organization with at least seven other GOST proteins in humans. Similarity to canonical GPCRs in the transmembrane region raises the question of whether TMEM87A or other GOST proteins are competent for GPCR signaling. While ligands have been proposed for some GOST proteins, specifically, neurostatin for GPR107 and L-lactate for GPR180, no definitive molecular evidence of G-protein signaling has been reported for any family member and GPR180 has been reported not to signal as a GPCR (*Bala-zova et al., 2021*; *Mosienko et al., 2018*; *Yosten et al., 2012*). The position of helix 8 in TMEM87A is sterically incompatible with G-protein or arrestin binding and predicted structures of other GOST proteins present similar steric blocks. Taken together, this raises the possibility that GOST proteins are structurally incompatible with canonical GPCR signaling.

If GOST proteins do not function as channels or GPCRs, what functional roles might they serve? Most GOST family proteins are relatively understudied, though previous work supports roles for six members in different aspects of protein transport (*Hirata et al., 2015*; *Shin et al., 2020*; *Tafesse et al., 2014*; *Carette et al., 2011*; *Elling et al., 2011*; *Zhou et al., 2014*; *Meisen et al., 2020*; *Dudek et al., 2020*; *Carette et al., 2009*; *Bänziger et al., 2006*; *Bartscherer et al., 2006*). The best studied is WLS, for which a large body of work describes an essential role in Wnt signaling. Wnts must be secreted from the cell to accomplish their signaling functions, but they are lipidated during maturation and therefore embedded within intracellular membranes. WLS facilitates Wnt transport and secretion by interacting simultaneously with hydrophilic Wnt extracellular regions (through its GOLD domain) and hydrophobic Wnt membrane-embedded regions (though the 7TM cavity). Structurally, TMEM87A and other GOST proteins display, or are predicted to display, key features suggesting they could serve analogous roles; they (i) have GOLD domains positioned to interact with extracellular soluble domains or client proteins, (ii) have large transmembrane cavities, and (iii) expose the cavities simultaneously to the extracellular side and the membrane. This raises the intriguing possibility that GOST family proteins could transport lipidated or otherwise hydrophobic secreted proteins through a mechanism analogous to Wnt binding and transport by WLS. In this scenario, the two domains of GOST proteins would perform a sort of 'coincidence detection' in which the GOLD domain binds solvent-exposed portions and the internal cavity binds membrane-buried portions of cargo molecules. Potential cargo for GOST proteins includes Wnts and other lipidated secreted proteins such as ghrelin and some

cytokines (*Port et al., 2011*; *Hang and Linder, 2011*; *Jiang et al., 2018*). Future studies are necessary to test the hypothesis that the GOST family proteins have common roles in transport of lipidated or membrane-embedded proteins.

## Materials and methods

### Cloning, expression, and purification

The coding sequence for *Homo sapiens* TMEM87A (Uniprot ID: Q8NBN3) was codon-optimized for expression in *Spodoptera frugiperda* cells (Sf9 cells) and synthesized (Integrated DNA Technologies). The sequence was cloned into a custom vector based on the pACEBAC1 backbone (MultiBac, Geneva Biotech) with an added C-terminal PreScission protease (PPX) cleavage site, linker sequence, superfolder GFP (sfGFP) and 7× His tag, generating a construct for expression of TMEM87A-SNS-LEVLFQGP-SRGGSGAAAGSGSGS-sfGFP-GSS-7×His. MultiBac cells were used to generate a Bacmid according to the manufacturer's instructions. Sf9 cells were cultured in ESF 921 medium (Expression Systems, these cells were not further authenticated or tested for Mycoplasma) and P1 virus was generated from cells transfected with FuGENE transfection reagent (Active Motif) according to the manufacturer's instructions. P2 virus was then generated by infecting cells at 2 million cells per ml with P1 virus at a multiplicity of infection of roughly 0.1, with infection monitored by fluorescence and harvested at 72 hr. P3 virus was generated in a similar manner to expand the viral stock. The P3 viral stock was then used to infect Sf9 cells at 2 million cells per mL at a multiplicity of infection of around 2–5. At 72 hr, infected cells containing expressed TMEM87A-sfGFP protein were collected by centrifugation at $1000 \times g$ for 10 min and frozen at −80°C. A cell pellet from 1 L culture was thawed and lysed by sonication in 100 mL buffer containing 50 mM Tris pH 8.0, 150 mM NaCl, 1 mM EDTA, and protease inhibitors (1 mM phenylmethylsulfonyl fluoride, 1 μM E64, 1 μg/mL pepstatin A, 10 μg/mL soy trypsin inhibitor, 1 μM benzamidine, 1 μg/mL aprotinin, and 1 μg/mL leupeptin). The membrane fraction was collected by centrifugation at $150,000 \times g$ for 45 min and homogenized with a cold Dounce homogenizer in 100 mL buffer containing 20 mM Tris pH 8.0, 150 mM NaCl, 1 mM EDTA, 1% n-dodecyl-b-D-maltopyranoside (DDM), 0.2% cholesteryl hemisuccinate (CHS), and protease inhibitors. Protein was extracted with gentle stirring for 2 hr at 4°C. The extraction mixture was centrifuged at $33,000 \times g$ for 45 min and the supernatant was bound to 5 mL Sepharose resin coupled to anti-GFP nanobody for 2 hr at 4°C. The resin was collected in a column and washed with 25 mL buffer 1 (20 mM Tris pH 8.0, 150 mM NaCl, 1 mM EDTA, 0.025% DDM, and 0.005% CHS), 50 mL buffer 2 (20 mM Tris pH 8.0, 500 mM NaCl, 1 mM EDTA, 0.025% DDM, and 0.005% CHS) and 25 mL buffer 1. The resin was then resuspended in 5 mL of buffer 1 with 0.5 mg PPX protease and rocked gently in the capped column overnight. Cleaved TMEM87A was eluted with an additional 8 mL buffer 1, spin concentrated to roughly 500 μL with Amicon Ultra spin concentrator 50 kDa cutoff (Millipore), and then loaded onto a Superose 6 increase column (GE Healthcare) on an NGC system (Bio-Rad) equilibrated in buffer 1. Peak fractions containing TMEM87A were then collected and spin concentrated before incorporation into proteoliposomes or nanodiscs.

### Nanodisc reconstitution

Freshly purified TMEM87A was reconstituted into MSP1D1 nanodiscs with a mixture of lipids (DOPE:POPS:POPC at a 2:1:1 mass ratio, Avanti) at a final molar ratio of 1:4:400 (TMEM87A:MSP-1D1:lipid mixture) (*Ritchie et al., 2009*). First, 20 mM solubilized lipids in nanodisc formation buffer (20 mM Tris pH 8.0, 150 mM NaCl, and 1 mM EDTA) was mixed with additional DDM detergent and TMEM87A. This solution was mixed at 4°C for 30 min before addition of purified MSP1D1. The solution with MSP1D1 was mixed at 4°C for 30 min before addition of 200 mg of Biobeads SM2. This mix was incubated at 4°C for 30 min before addition of another 200 mg of Biobeads. This final mixture was then gently tumbled at 4 °C overnight (roughly 12 hr). Supernatant was cleared of beads by letting large beads settle and carefully removing liquid with a pipette. Sample was spun for 10 min at $21,000 \times g$ before loading onto a Superose 6 increase column in buffer containing 20 mM Tris pH 8.0, 150 mM NaCl. Peak fractions corresponding to TMEM87A in MSP1D1 were collected, 50 kDa cutoff spin concentrated and used for grid preparation. MSP1D1 was prepared as described without cleavage of the His-tag.

## EM sample preparation and data collection

TMEM87A in MSP1D1 nanodiscs was centrifuged at 21,000×*g* for 5 min at 4°C. A 3 μL sample was applied to holey carbon, 300 mesh R1.2/1.3 gold grids (Quantifoil, Großlöbichau, Germany) that were freshly glow discharged for 25 s. Sample was incubated for 5 s at 4 °C and 100% humidity prior to blotting with Whatman #1 filter paper for 3 s at blot force 1 and plunge-freezing in liquid ethane cooled by liquid nitrogen using an FEI Mark IV Vitrobot (FEI/Thermo Fisher Scientific, USA). Grids were clipped and transferred to an FEI Talos Arctica electron microscope operated at 200 kV. Fifty frame movies were recorded on a Gatan K3 Summit direct electron detector in super-resolution counting mode with pixel size of 0.5575 Å. The electron dose rate was 8.849 e$^-$ Å$^2$ s$^{-1}$ and the total dose was 50.0 e$^-$ Å$^2$. Nine movies were collected around a central hole position with image shift and defocus was varied from −0.6 to −1.8 μm through SerialEM (*Mastronarde, 2005*).

## Cryo-EM data processing

7060 micrograph movies were motion-correction with dose-weighting using RELION3.1's implementation of MotionCor2 and 'binned' 2× from super-resolution to the physical pixel size (*Zheng et al., 2017*; *Zivanov et al., 2018*; *Zivanov et al., 2019*). CTFFIND-4.1 was then used to estimate the contrast transfer function (CTF) (*Rohou and Grigorieff, 2015*). Micrographs with a CTF maximum estimated resolution worse than 4 Å were discarded, yielding 5898 micrographs. Particle images were auto-picked first with RELION3.1's Laplacian-of-Gaussian filter and, following initial clean-up and 2D classification, templated-based auto-picking was performed, yielding 2,674,406 particles. Two parallel paths were pursued to identify 'good' particles from this set. In one path, iterative 2D classification in RELION yielded 495,963 particles and 2D classification, ab initio reconstruction, heterogeneous refinement, and nonuniform refinement (in CryoSPARC, *Punjani et al., 2017*) yielded 185,958 particles and a 5.7 Å reconstruction. In a second path, iterative 2D classification, ab initio reconstruction, heterogeneous refinement, and nonuniform refinement (in CryoSPARC) yielded 100,165 particles and a 5.3 Å reconstruction. Particles from each path were combined, pruned of duplicates within a 100 Å cutoff, and subjected to 2D classification and 3D refinement in RELION yielding 262,901 particles and a 5.9 Å reconstruction. Particles were Bayesian polished in RELION and subjected to iterative rounds of ab initio reconstruction and nonuniform refinement in CryoSPARC yielding 112,707 particles and a 4.8 Å reconstruction. These particle positions were used for training in Topaz, and the resulting Topaz model used to repick 1,468,201 particles (*Bepler et al., 2019*). Iterative 2D classification in RELION and CryoSPARC resulted in 230,350 Topaz-picked particles which were merged with the training particle set, pruned of duplicates within a 100 Å cutoff, and re-centered, yielding 322,094 particles. Iterative ab initio reconstruction, 2D classification, and nonuniform refinement in CryoSPARC yielded 138,217 particles. This final set of particles was subjected to 3D refinement and Bayesian polishing in RELION and ab initio and nonuniform refinement in CryoSPARC until no further improvements in map quality were observed (*Punjani et al., 2020*). The output of the best nonuniform refinement was used for particle subtraction in RELION, after which final 3D refinement and postprocessing jobs were performed. This final refinement was input to Phenix resolve density modification to generate an additional map used during modeling (*Terwilliger et al., 2020*). Throughout processing, initial resolution and dynamic mask nonuniform parameters were adjusted empirically to yield the best performing refinement. UCSF pyem was used for conversion of files from cryoSPARC to RELION formats (*Asarnow et al., 2019*).

## Model building and refinement

The final relion postprocessed map and Phenix density modified map were used for modeling. The AlphaFold2 TMEM87A model was rigid body fit to the density in ChimeraX and used as a foundation for manual model building in Coot (*Goddard et al., 2018*; *Emsley et al., 2010*). The model was real space refined in Phenix and assessed for proper stereochemistry and geometry using Molprobity (*Liebschner et al., 2019*; *Williams et al., 2018*). FSCs calculated in Phenix mtriage. Structures were analyzed and figures were prepared with CASTp, DALI, ChimeraX, JalView, Prism 8, Python GNU Image Manipulation Program, and Adobe Photoshop and Illustrator software (*Holm, 2020*; *Tian et al., 2018*; *Waterhouse et al., 2009*). Consurf was used to map conservation onto the structure surfaces using an alignment of sequences determined using SHOOT (*Ashkenazy et al., 2016*; *Emms*

*and Kelly, 2022*). Electrostatic potentials calculated in ChimeraX. CASTp cavity calculation output was converted from JSON to PDB using a custom python script.

## Proteoliposome reconstitution

For proteoliposome patching experiments, we incorporated protein into lipids and generated proteoliposome blisters for patch recordings using dehydration and rehydration as described previously (*Del Mármol et al., 2018*) with the following modifications. TMEM87A in buffer 1 was exchanged into soybean l-α-phosphatidylcholine (Soy PC, MillaporeSigma) with the addition of Biobeads SM2 (Bio-Rad) and an hour incubation at a protein:lipid ratio of 1:10 corresponding to 0.4 mg purified TMEM87A and 4 mg of Soy PC lipid or 1:50 in buffer (5 mM HEPES pH 7.2, 200 mM KCl). TRAAK control proteoliposomes were prepared at 1:50 as described previously. Control liposomes were prepared from the same lipid and protocol with protein replaced with buffer 1.

## Electrophysiology

Proteoliposomes were thawed and dispensed in 0.5–1 µL drops on a 35 mm glass-bottom dish. The drops were dried in a vacuum chamber in the dark overnight. Proteoliposomes were rehydrated with 20 µL buffer (5 mM HEPES pH 7.2, 200 mM KCl). Each cake was firstly covered with a buffer drop and then let surface tension connect drops. Rehydrating proteoliposomes were placed within a humid chamber at 4°C before patching. Recordings were made at room temperature using Clampex v.10.7 data acquisition software (as part of the pClamp v.10.7 suite) with an Axopatch 200B Patch Clamp amplifier and Digidata 1550B digitizer (Molecular Devices) at a bandwidth of 1 kHz and digitized at 500 kHz. A pressure clamp (ALA Scientific) was used to form seals. Pipette solution was 10 mM HEPES pH 7.2, 150 mM KCl, 3 mM $MgCl_2$, and 5 mM EGTA. Bath solution was 10 mM HEPES pH 7.3, 135 mM NaCl, 15 mM KCl, 1 mM $CaCl_2$, and 3 mM $MgCl_2$. Borosilicate glass pipettes were pulled and polished to a resistance of 2–5 MΩ when filled with pipette solution.

## Acknowledgements

The authors thank J Remis, D Toso, and P Tobias for microscope and computational support at the Cal-Cryo facility of UC Berkeley. The authors thank all members of the Brohawn lab for helpful discussions and critical feedback on the project. S.G.B. is a New York Stem Cell Foundation-Robertson Neuroscience Investigator. This work was funded by the New York Stem Cell Foundation; NIGMS grant GM123496; a McKnight Foundation Scholar Award; a Sloan Research Fellowship; and a Winkler Family Scholar Award (to S.G.B.)

## Additional information

### Funding

| Funder | Grant reference number | Author |
|---|---|---|
| New York Stem Cell Foundation | | Stephen G Brohawn |
| National Institute of General Medical Sciences | GM123496 | Stephen G Brohawn |
| McKnight Foundation | | Stephen G Brohawn |

The funders had no role in study design, data collection and interpretation, or the decision to submit the work for publication.

### Author contributions

Christopher M Hoel, C.M.H. processed cryo-EM data and modeled the structure; Lin Zhang, L.Z. cloned; Stephen G Brohawn, S.G.B. secured funding and supervised the project

### Author ORCIDs

Christopher M Hoel http://orcid.org/0000-0002-1344-0780
Lin Zhang http://orcid.org/0000-0002-8132-7755

Stephen G Brohawn http://orcid.org/0000-0001-6768-3406

**Decision letter and Author response**
Decision letter https://doi.org/10.7554/eLife.81704.sa1
Author response https://doi.org/10.7554/eLife.81704.sa2

## Additional files

### Supplementary files
• Supplementary file 1. Cryo-EM data collection, processing, refinement, and modeling statistics.
• MDAR checklist

### Data availability
All data associated with this study ware publicly available. The TMEM87A model is in the Protein Data Bank (PDB) under 8CTJ, the final maps are in the Electron Microscopy Data Bank (EMDB) under EMD-26992, and the original micrograph movies and final particle stack are in the Electron Microscopy Public Image Archive (EMPIAR) under EMPIAR-11045.

The following datasets were generated:

| Author(s) | Year | Dataset title | Dataset URL | Database and Identifier |
|---|---|---|---|---|
| Hoel CM, Zhang L, Brohawn SG | 2022 | Cryo-EM structure of TMEM87A | https://www.rcsb.org/structure/8CTJ | RCSB Protein Data Bank, 8CTJ |
| Hoel CM, Zhang L, Brohawn SG | 2022 | Cryo-EM structure of TMEM87A | https://www.ebi.ac.uk/emdb/EMD-26992 | Electron Microscopy Data Bank (EMDB) under, EMD-26992 |
| Hoel CM, Zhang L, Brohawn SG | 2022 | Cryo-EM structure of the GOLD-domain seven-transmembrane protein TMEM87A | https://www.ebi.ac.uk/empiar/EMPIAR-11045/ | Electron Microscopy Public Image Archive (EMPIAR) under, EMPIAR-11045 |

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
