## [Editor Report]

Your work addresses the mechanisms of action of the transmembrane proteins TMEM87A and TMEM87B, which are thought to play a role in protein transport but have been implicated in other processes as well, such as signaling and acting as mechanosensitive ion channels. The study represents an important advance in the understanding of this poorly characterized family of proteins.

---

## [Decision Letter]

**Decision letter after peer review:**

Thank you for submitting your article "Structure of the GOLD-domain seven-transmembrane helix protein family member TMEM87A" for consideration by *eLife*. Your article has been reviewed by 4 peer reviewers, one of whom is a member of our Board of Reviewing Editors, and the evaluation has been overseen by Richard Aldrich as the Senior Editor. The reviewers have opted to remain anonymous.

Essential revisions:

The reviewers are not asking for additional experiments but they make a number of recommendations with respect to the interpretation of the data, as well as the inclusion of additional data. For example, reviewer #2 suggests that it would be helpful to show the EM density map in Figure 1, to help the readers to understand the limitations of the structure. Reviewer # 3 asks why the structure is so low in resolution and more information as well.

We suggest that you submit a revised version of your manuscript addressing these and other questions from the reviewers.

*Reviewer #1 (Recommendations for the authors):*

The work would be of much wider interest if a target molecule would be known.

*Reviewer #2 (Recommendations for the authors):*

While the work is well done, I have several suggestions that could improve the interpretability and accessibility of the study for a general audience.

1. In the introduction, the authors mention previous studies that suggested that TMEM87s may participate in mechanosensation. In contrast, in this work, no mechanically activated currents could be observed from proteoliposomes containing purified proteins. Were the authors able to replicate any of these findings in a cellular context? Would it be possible to speculate on the differences between the in vitro and in cell environments and could that inform on the roles of TMEM87s?

2. Because of the low resolution of the reconstruction, it would be helpful to show the EM density map in Figure 1. It would help the readers to understand the limitations of the structure.

3. Continuing from point 2, the authors should take care with the assignment of the non-protein density in the cavity as a lipid molecule. The text should be toned down to note that while there is density, it could also be attributed to noise and the model shown in Figure 5c is highly speculative.

*Reviewer #3 (Recommendations for the authors):*

The major issue with this manuscript lies in the lack of data. A low resolution cryo-EM structure has been solved and compared to other structures and models to generate the hypothesis that this protein may have roles in trafficking membrane-associated cargo, but the experiments to demonstrate these have not been performed thus rendering this manuscript incomplete.

It is unclear as to why the structure is so low in resolution as very little information is provided re the collection and processing (e.g., micrograph number, particle number, box size) and it seems that the processing was not exhaustive – did the authors try local refinement, 3D classification without alignment, 3DVA? Without the details for the collection and processing it's hard to tell what the cause of this is, but in my opinion, it could be overcome by further processing, collection on a 300 keV microscope (which the authors should have access to through national facilities), and if necessary, fiducials to increase the size of the particle.

*Reviewer #4 (Recommendations for the authors):*

1. Although no current was recorded in patched membrane, this negative result cannot formally exclude the possibility that TMEM87A may be part of an ion channel. The caveat should be discussed.

2. Is helix 8 conserved across species in TMEM87A? Are the interactions between helix 8 and the intracellular surface of TM domain conserved? If so, this might help to strengthen the argument that helix 8 may block G-protein or Arrestin coupling.

3. Is it possible to carry out some functional characterizations? For example, rescuing defective retrograde traffic (as mentioned in the introduction), if this assay is feasible. This might help to test whether its GOLD domain (and possibly cavity) is functionally important as suggested in the manuscript (based on comparison with WLS).

4. It will be useful to present a more detailed workflow of the data processing in the supplementary figure.

---

## [Author Response]

Essential revisions:The reviewers are not asking for additional experiments but they make a number of recommendations with respect to the interpretation of the data, as well as the inclusion of additional data. For example, reviewer #2 suggests that it would be helpful to show the EM density map in Figure 1, to help the readers to understand the limitations of the structure. Reviewer # 3 asks why the structure is so low in resolution and more information as well.We suggest that you submit a revised version of your manuscript addressing these and other questions from the reviewers.

We have addressed these concerns in the revised manuscript. Figure 1 has been updated to include a map image and cryo-EM data processing is more fully described through additional methods text and a new processing pipeline figure (Figure 1 —figure supplement 3).

Reviewer #1 (Recommendations for the authors):The work would be of much wider interest if a target molecule would be known.

We agree, but think extensive experimentation required to identify putative target molecules is better suited to future studies.

Reviewer #2 (Recommendations for the authors):While the work is well done, I have several suggestions that could improve the interpretability and accessibility of the study for a general audience.1. In the introduction, the authors mention previous studies that suggested that TMEM87s may participate in mechanosensation. In contrast, in this work, no mechanically activated currents could be observed from proteoliposomes containing purified proteins. Were the authors able to replicate any of these findings in a cellular context? Would it be possible to speculate on the differences between the in vitro and in cell environments and could that inform on the roles of TMEM87s?

We have added text in the discussion related to this point. We have not attempted to replicate previously reported experiments. We now more fully describe the reconstituted recording system in the text to make the limitations of this experiment clear. While our functional data and the structural relationship between TMEM87A and WLS suggests that TMEM87A plays other functional roles and may rather indirectly impact channel activity, we cannot fully exclude the possibility TMEM87A contributes more directly to channel activity in a cellular context. Future work will be required to answer this question definitively.

2. Because of the low resolution of the reconstruction, it would be helpful to show the EM density map in Figure 1. It would help the readers to understand the limitations of the structure.

Thank you for this suggestion. We have added a map image to Figure 1a.

3. Continuing from point 2, the authors should take care with the assignment of the non-protein density in the cavity as a lipid molecule. The text should be toned down to note that while there is density, it could also be attributed to noise and the model shown in Figure 5c is highly speculative.

We agree and have edited the text to emphasize that this is a speculative assignment at this resolution and is based on location and relationship to WLS in addition to EM density.

Reviewer #3 (Recommendations for the authors):The major issue with this manuscript lies in the lack of data. A low resolution cryo-EM structure has been solved and compared to other structures and models to generate the hypothesis that this protein may have roles in trafficking membrane-associated cargo, but the experiments to demonstrate these have not been performed thus rendering this manuscript incomplete.

We agree addressing the hypothesis raised from the structural work presented here is the most important next step, but think these experiments are better left for future studies.

It is unclear as to why the structure is so low in resolution as very little information is provided re the collection and processing (e.g., micrograph number, particle number, box size) and it seems that the processing was not exhaustive – did the authors try local refinement, 3D classification without alignment, 3DVA? Without the details for the collection and processing it's hard to tell what the cause of this is, but in my opinion, it could be overcome by further processing, collection on a 300 keV microscope (which the authors should have access to through national facilities), and if necessary, fiducials to increase the size of the particle.

We have added detail to the methods and cryo-EM data processing pipeline figure related to this point. We speculate that resolution is limited by the presence of residual conformational heterogeneity among particles we are unable to classify and/or data quality and quantity. The resolution achieved was sufficient for defining the architecture of TMEM87A and facilitating the discovery of its relationship to WLS and other GOST proteins, so we prefer to pursue additional data collection in future work. We note cryo-EM structure determination of such small membrane proteins (TMEM87A is 63 kDa) remains technically challenging. To our knowledge, no cryo-EM structure of a comparable monomeric, apo-7TM protein has been resolved to higher resolution (a 73 kDa CGRP receptor-RAMP1 complex reconstruction reached 3.15Å resolution (Josephs et al., *Science* 2021, PDB: 7KNT)). We tested many alternative and additional strategies including the suggested local refinement (with various masks), 3D classification without alignment, and 3D variability analysis approaches, but were unable to further improve resolution of the reconstruction. All data are publicly available (including raw micrographs) so alternative approaches can be evaluated in future.

Reviewer #4 (Recommendations for the authors):1. Although no current was recorded in patched membrane, this negative result cannot formally exclude the possibility that TMEM87A may be part of an ion channel. The caveat should be discussed.

We fully agree and further emphasize this point in the discussion.

2. Is helix 8 conserved across species in TMEM87A? Are the interactions between helix 8 and the intracellular surface of TM domain conserved? If so, this might help to strengthen the argument that helix 8 may block G-protein or Arrestin coupling.

We thank the reviewer for this suggestion. Indeed, both helix 8 and the intracellular surface it interacts with are highly conserved. This is now discussed in the text and shown in a new figure panel (Figure 2 —figure supplement 3e).

3. Is it possible to carry out some functional characterizations? For example, rescuing defective retrograde traffic (as mentioned in the introduction), if this assay is feasible. This might help to test whether its GOLD domain (and possibly cavity) is functionally important as suggested in the manuscript (based on comparison with WLS).

We think these studies, ideally involving unbiased identification of putative cargo or screening for functional roles, are better suited for future studies.

4. It will be useful to present a more detailed workflow of the data processing in the supplementary figure.

A more detailed data processing pipeline is now shown in Figure 1 —figure supplement 3. Additional text has also been added to the methods.